# Quarkonium Phenomenology from a Generalised Gauss Law

**David Lafferty [1],\*  and Alexander Rothkopf [2]**

[1]    Institute for Theoretical Physics, Heidelberg University, Philosophenweg 16, 69120 Heidelberg, Germany
[2]    Faculty of Science and Technology, University of Stavanger, 4021 Stavanger, Norway;
       alexander.rothkopf@uis.no
\*    Correspondence: lafferty@thphys.uni-heidelberg.de

**Abstract:** We present an improved analytic parametrisation of the complex in-medium heavy quark potential derived rigorously from the generalised Gauss law. To this end we combine in a self-consistent manner a non-perturbative vacuum potential with a weak-coupling description of the QCD medium. The resulting Gauss-law parametrisation is able to reproduce full lattice QCD data by using only a single temperature dependent parameter, the Debye mass $m_D$. Using this parametrisation we model the in-medium potential at finite baryo-chemical potential, which allows us to estimate the $\Psi'/J/\Psi$ ratio in heavy-ion collisions at different beam energies.

**Keywords:** quarkonium; heavy-quark potential; heavy-ion collisions; quarkonium phenomenology

## 1. Introduction

The study of heavy-quarkonium—the bound states of a heavy quark anti-quark pair—has become a central tenet in our understanding of strongly interacting matter under extreme conditions in the context of heavy-ion collisions. Experimentally, the decay of heavy quarkonia into di-leptons leaves a clean signal that allows the probing of different stages of the quark gluon plasma (QGP) and ensures the continued importance of heavy quarkonium measurements at future accelerators [1]. On the theory side, the heavy masses of the constituent quarks permits the use of effective field theories (EFTs) to simplify the description of heavy quarkonium behaviour [2]. This powerful framework has led to considerable progress both in direct lattice QCD studies of equilibrated quarkonium as well as in real-time descriptions of their non-equilibrium evolution. The formulation of EFTs relies on a separation of scales inherent to the heavy-quark system, $m_Q \ll m_Q v \ll m_Q v^2$ with $m_Q$ the heavy-quark mass and $v$ its typical velocity, denoted respectively as hard, soft, and ultra-soft. Two additional scales are present, namely the characteristic scale of quantum fluctuations $\Lambda_{\mathrm{QCD}}$ and of thermal fluctuations $T$. Integrating out the hard scale $\sim m_Q$ from the full Quantum ChromoDynamics (QCD) Lagrangian leaves Non-Relativistic QCD (NRQCD) given in terms of non-relativistic Pauli spinor fields; this can be achieved non-perturbatively. Further integrating out the soft scale $\sim m_Q v$ results in Potential Non-Relativistic QCD (pNRQCD), where the potential governing the quarkonium dynamics enters as a matching coefficient. While the perturbative derivation of pNRQCD has been successfully completed, its non-perturbative definition is still an active field of research.

In the static limit, the EFT-based definition of such a potential has been suggested based on the real-time evolution on the QCD Wilson loop [3]:

$$V(r) = \lim_{t \to \infty} \frac{i \partial_t W_\square(r,t)}{W_\square(r,t)}. \tag{1}$$

The evaluation of Equation (1) in hard thermal loop (HTL) resummed perturbation theory has demonstrated that this potential is a complex quantity [4]. In addition to the well-known Debye screening in the real part, an imaginary part arises owing to Landau damping or gluo-dissociation, depending on the hierarchy of scales present [5]. At high temperatures the former dominates and the potential reads:

$$V_{HTL}(r) = -\tilde{\alpha}_s \left[ m_D + \frac{e^{-m_D r}}{r} + iT\phi(m_D r) \right] + \mathcal{O}(g^4), \quad \phi(x) = 2 \int_0^\infty dz \, \frac{z}{(z^2+1)^2} \left( 1 - \frac{\sin(xz)}{xz} \right).$$ (2)

Here $\tilde{\alpha}_s = C_F g^2 / 4\pi$ is the rescaled strong coupling constant. It should be emphasised that this potential does not govern the evolution of the bound state wavefunction; instead it evolves the correlator of unequal time wavefunctions. The question of how this potential can be related to the evolution of the wavefunction itself is an active field of research—an open-quantum-systems approach appears to be promising in this regard (see, e.g., [6]).

Significant progress has been made in understanding the equilibrated properties of heavy quarkonium by extracting the heavy quark potential directly from lattice QCD simulations. These works have confirmed that at low temperatures the potential closely resembles the Cornell form [7],

$$V^{\text{vac}}(r) = -\frac{\tilde{\alpha}_s}{r} + \sigma r + c,$$ (3)

where $\sigma$ denotes the string-tension and $c$ an additive constant. Equation (3) already captures the two most prominent features of QCD, namely asymptotic freedom via the running coupling at small distances and confinement via the non-perturbative linear rise. At finite temperature, the same extraction procedure reveals a weakening of the real part as one moves into the deconfined phase, as well as an imaginary part persisting beyond the QCD pseudo-critical temperature. In order to employ these numerical results in computations of quarkonium spectral functions, which inform us of the in-medium properties, we require an accurate analytic parametrisation of the in-medium heavy quark potential—in particular that holds at the lower and more phenomenologically relevant temperatures below the strict validity range of HTL perturbation theory.

To this end, in this contribution we improve upon the work of [8] and utilise the generalised Gauss law to reproduce the in-medium heavy quark potential. The non-perturbative vacuum bound state is described by the Cornell potential in Equation (3) and will be inserted into a weakly coupled deconfined medium characterised by the HTL in-medium permittivity. Taking into account string breaking, we are able to derive expressions for Re$V$ and Im$V$ with a closed and simple functional form. This parametrisation captures the in-medium behaviour of the real and imaginary parts of the lattice-QCD-calculated potential very well, based on a single temperature dependent parameter—the Debye mass $m_D$. Our new derivation overcomes the main technical limitation of the previous work, namely an ad-hoc assumption about the functional form of the real-space in-medium permittivity.

## 2. The Gauss Law Potential Model

### 2.1. A Novel Formulation

The central idea of this approach is to calculate the in-medium modification to the Coulombic and string-like parts of the Cornell potential given in Equation (3). In linear response theory, the electric potential at finite temperature is obtained from its vacuum counterpart via a division in momentum-space by the static dielectric constant [9]:

$$V(\mathbf{p}) = \frac{V^{\text{vac}}(\mathbf{p})}{\varepsilon(\mathbf{p}, m_D)}.$$ (4)

The permittivity, defined as an appropriate limit of the real-time in-medium gluon propagator, will encode the medium effects. Equation (4) does not rely on a weak-coupling approximation and

remains valid so long as the vacuum field is weak enough to justify the linear response ansatz. The real space equivalent via the convolution theorem is

$$V(\mathbf{r}) = \left( V^{\text{vac}} * \varepsilon^{-1} \right)(\mathbf{r}),\tag{5}$$

where '$*$' represents the convolution. We now consider the other main building block of our approach, the generalised Gauss law,

$$\nabla \cdot \left( \frac{\mathbf{E}^{\text{vac}}}{r^{a+1}} \right) = 4\pi q \delta(\mathbf{r}),\tag{6}$$

which holds for electric fields of the form $\mathbf{E}^{\text{vac}}(r) = -\nabla V^{\text{vac}}(r) = q r^{a-1} \hat{r}$. This reduces to the well-known Coulombic potential for $a = -1, q = \tilde{\alpha}_s$ while the linearly rising string case corresponds to $a = 1, q = \sigma$. For a general $a$,

$$-\frac{1}{r^{a+1}}\nabla^2 V^{\text{vac}}(r) + \frac{1+a}{r^{a+2}}\nabla V^{\text{vac}}(r) = 4\pi q \delta(\mathbf{r}).\tag{7}$$

Denoting the differential operator on the left-hand-side above as $\mathcal{G}_a$ and applying it to Equation (5), the general integral expressions for each term in the in-medium heavy-quark potential are deduced:

$$\mathcal{G}_a\left[V(r)\right] = \mathcal{G}_a \int \mathrm{d}^3y \left( V^{\text{vac}}(r-y)\varepsilon^{-1}(y) \right) = 4\pi q \left( \delta * \varepsilon^{-1} \right)(r) = 4\pi q\, \varepsilon^{-1}(r, m_D).\tag{8}$$

Here we have used Equation (7) and that the convolution commutes with $\mathcal{G}_a$. For the Coulombic and string cases respectively, this gives

$$-\nabla^2 V_C(r) = 4\pi\tilde{\alpha}_s\, \varepsilon^{-1}(r, m_D), \qquad -\frac{1}{r^2}\frac{\mathrm{d}^2 V_S(r)}{\mathrm{d}r^2} = 4\pi\sigma\, \varepsilon^{-1}(r, m_D).\tag{9}$$

From the perturbative HTL expression in momentum-space [10],

$$\varepsilon^{-1}(p, m_D) = \frac{p^2}{p^2 + m_D^2} - i\pi T \frac{pm_D^2}{\left(p^2 + m_D^2\right)^2},\tag{10}$$

the expression for the coordinate space in-medium permittivity is obtained by inverse Fourier transform. Now, using Equation (10) to solve for the in-medium modified Coulombic part of the potential, we find that our ansatz reproduces the HTL result

$$\mathrm{Re}V_C(r) = -\tilde{\alpha}_s\left[ m_D + \frac{e^{-m_D r}}{r} \right], \qquad \mathrm{Im}V_C(r) = -\tilde{\alpha}_s\left[ iT\phi(m_D r) \right],\tag{11}$$

with $\phi$ as defined in Equation (2). The next step is to turn to the string part, for which the formal solution can be immediately written down as

$$V_S(r) = c_0 + c_1 r - 4\pi\sigma \int_0^r \mathrm{d}r' \int_0^{r'} \mathrm{d}r'' r''^2 \varepsilon^{-1}\left(r'', m_D\right).\tag{12}$$

The constants $c_0$ and $c_1$ will be chosen to ensure the physically motivated boundary conditions $\mathrm{Re}V_S(r)|_{r=0} = 0$, $\mathrm{Im}V_S(r)|_{r=0} = 0$ and $\partial_r\mathrm{Im}V_S(r)|_{r=0} = 0$. This leads to the following analytical form:

$$\mathrm{Re}V_S(r) = \frac{2\sigma}{m_D} - \frac{e^{-m_D r}\left(2 + m_D r\right)\sigma}{m_D}, \qquad \mathrm{Im}V_S(r) = \frac{\sqrt{\pi}}{4}m_D T\sigma\, r^3\, G_{2,4}^{2,2}\left( \left. \begin{matrix} -\frac{1}{2}, -\frac{1}{2} \\ \frac{1}{2}, \frac{1}{2}, -\frac{3}{2}, -1 \end{matrix} \right| \frac{1}{4}m_D^2 r^2 \right),\tag{13}$$

where $G$ denotes the Meijer-G function. In the real parts the short distance limit $r \to 0$ recovers the Cornell potential as does the zero temperature limit $m_D \to 0$. At large distances $\mathrm{Re}V_C(r)$ displays an

exponential decay $\sim e^{-m_D r}$ (i.e., Debye screening) while $\mathrm{Im}V_C(r)$ asymptotes to a constant which is expected for Landau damping. Only the imaginary string part in Equation (13), at first sight appears problematic as it diverges logarithmically at large $r$. We argue that this is a manifestation of the absence of an explicit string breaking in the original vacuum Cornell potential.

In the preceding computation the explicit expression for $\mathrm{Im}V_S$ can be written, after substituting the imaginary part of Equation (10) into Equation (12) and performing the angular integration of the inverse Fourier transform, as follows:

$$\mathrm{Im}V_S(r) = c_0 + c_1 r + 2T\sigma m_D^2 \int_0^r dr' \int_0^{r'} dr'' \, r''^2 \int_0^\infty dp \, p^2 \, \frac{\sin(pr'')}{pr''} \, p^2 \, \frac{1}{p\left(p^2 + m_D^2\right)^2}. \tag{14}$$

We have arranged the momentum factors as above to make clear their different origins: the first term ($p^2$) arises from integrating in spherical coordinates and the second ($\mathrm{sinc}(pr'')$) after completing the polar integration. The last two terms are contributions from the in-medium permittivity. It is the $1/p$ factor here that we identify as causing the weak infrared divergence. In order to regularise, we modify this term as

$$\frac{1}{p\left(p^2 + m_D^2\right)^2} \rightarrow \frac{1}{\sqrt{p^2 + \Delta^2}\left(p^2 + m_D^2\right)^2}, \tag{15}$$

where $\Delta$ will be a suitably chosen regularisation scale. In Equation (14) the spatial integrals can be carried out analytically, which combined with the regularisation above gives our new definition of the string imaginary part:

$$\mathrm{Im}V_S(r) = 2T\sigma m_D^2 \int_0^\infty dp \, \frac{2 - 2\cos(pr) - pr\sin(pr)}{\sqrt{p^2 + \Delta^2}\left(p^2 + m_D^2\right)^2}, \tag{16}$$

after imposing the boundary conditions stated above Equation (13). The only remaining step is to determine the regularisation scale $\Delta$. To do so, note that if we rescale momentum $p \rightarrow p/m_D$ and slightly rearrange, Equation (16) takes on a suggestive form:

$$\mathrm{Im}V_S(r) = \frac{\sigma T}{m_D^2}\chi(m_D r), \quad \chi(x) = 2\int_0^\infty dp \, \frac{2 - 2\cos(px) - px\sin(px)}{\sqrt{p^2 + \Delta_D^2}\left(p^2 + 1\right)^2}, \tag{17}$$

with $\Delta_D = \Delta/m_D$. That is, we can express $\mathrm{Im}V_S(r)$ using a temperature dependent prefactor with dimensions of energy, multiplied by a dimensionless momentum integral. This is very similar to the Coulombic expression, where the integral asymptotes to unity in the limit $r \rightarrow \infty$. We thus impose the same condition for the string part. This procedure also recovers the correct behaviour at large $T$ (large $m_D$), i.e., the string contribution to the imaginary part diminishes until the HTL result is recovered. The value of the regularisation parameter $\Delta_D$ can be computed numerically. Furthermore, since it is expressed in terms of the Debye mass it remains constant and the computation need only be performed once. It is found that $\Delta_D = \Delta/m_D \simeq 3.0369$ gives $\chi(\infty) \simeq 1$ and thus Equation (17) represents the final closed form of a physically consistent in-medium string imaginary part.

## 2.2. Vetting with Lattice QCD Data

The most important benchmark for any description of the in-medium heavy quark potential is its ability to reproduce the non-perturbative lattice QCD results. This vetting process is carried out here against potential values [11] calculated on finite temperature ensembles generated by the HotQCD collaboration on $48^3 \times 12$ lattices with $N_f = 2 + 1$ flavours of dynamical light quarks discretised with the asqtad action [12]. The pion mass on these lattices is $m_\pi \approx 300$ MeV and the QCD transition temperature is $T_C \approx 175$ MeV.

Following the steps in [13], we first calibrate the vacuum parameters by fitting the Cornell potential to the two low-temperature ensembles included in the lattice dataset. As in that study, the

Cornell ansatz gives an excellent fit. The entire temperature dependence in our parametrisation then enters only via the Debye mass $m_D$, which will be fit using only the real part. The imaginary data points can be used as a cross-check. Note that since the heavy quark potential is a generic quantity that is unspecific to either of the heavy quark families, this fit need only be performed once.

The results are shown in Figure 1. From the left panel we see that the Gauss law parametrisation provides an excellent fit, capturing the behaviour of the non-perturbative data points from the Coulombic region at small $r$ through the intermediate region and up to the screening regime at high temperature and large distances. Furthermore, the central panel shows a good agreement between the Gauss law predictions and corresponding tentative values of the imaginary part extracted from the lattice. The predicted values lie within the considerable errors of the lattice $\mathrm{Im}V_S(r)$ for all but the lowest temperature. We observe that the imaginary part from the Gauss law rises more steeply with increasing temperature but the asymptotic value at large distances behaves non-monotonously, reflecting the competing Coulombic and string parts. The best fit values of $m_D$ are shown in the right panel. We conclude that our novel parametrisation captures the relevant physics encoded within the non-perturbative in-medium potential.

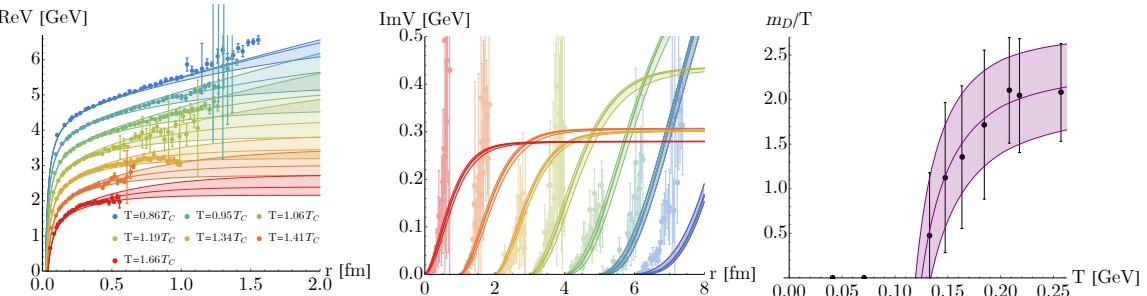

**Figure 1.** Gauss-law parametrisation and the lattice QCD potential. (**left**) Real part (symbols) and best fit results (solid lines). (**centre**) Tentative imaginary part (symbols) and the Gauss-law prediction (solid lines). Errorbands from uncertainty in both the $T > 0$ fit and the vacuum parameters. (**right**) Best fit values of the Debye mass and interpolation.

## 3. Phenomenology

### 3.1. Spectral Functions at Finite Temperature

The next natural step is to employ our validated Gauss law potential model in a realistic investigation of heavy quarkonium in-medium behaviour. As we have calibrated the Debye mass temperature dependence against lattice data with an unphysical pion mass, we first must carry out a continuum extrapolation. Since this has not been rigorously achieved so far we resort to using continuum corrections as outlined in detail in [13]. The outcome is a set of phenomenological vacuum parameters for the Cornell potential, which in our case read

$$\tilde{\alpha}_s = 0.513 \pm 0.0024 \,\mathrm{GeV}, \quad \sqrt{\sigma} = 0.412 \pm 0.0041 \,\mathrm{GeV}, \quad c = -0.161 \pm 0.0025 \,\mathrm{GeV}, \qquad (18)$$

to be used in conjunction with a "fit" of the charm mass $m_c^{\mathrm{fit}} = 1.4692$ GeV. The continuum corrected values for the Debye mass parameter are interpolated via the HTL inspired ansatz

$$m_D(T) = Tg(\Lambda) \sqrt{\frac{N_c}{3} + \frac{N_f}{6}} + \frac{N_c Tg(\Lambda)^2}{4\pi} \log\left(\frac{1}{g(\Lambda)} \sqrt{\frac{N_c}{3} + \frac{N_f}{6}}\right) + \kappa_1 Tg(\Lambda)^2 + \kappa_2 Tg(\Lambda)^3 . \quad (19)$$

Here, the first and second term respectively are the leading order perturbative result plus logarithmic correction in $SU(N_c)$ with $N_f$ fermions, $m_{u,d} = 0$, and at zero baryon chemical potential. $\kappa_1$ and $\kappa_2$ absorb the non-perturbative corrections, which in our case take the values $\kappa_1 = 0.686 \pm 0.221$

and $\kappa_2 = -0.317 \pm 0.052$. The resulting interpolation for $m_D$ is shown as the purple band in the right panel of Figure 1.

With these corrections in place, we may now calculate realistic quarkonium spectral functions at finite temperature by solving the appropriate Schrödinger equation using the Fourier space method as described in [14].

In Figure 2 we show the results for S-wave charmonium states, which exhibit the characteristic broadening of in-medium peaks and their shifts to lower frequencies. This corresponds to the in-medium state being lighter than the vacuum state, while at the same time being less strongly bound. The in-medium modification is shown quantitatively in Figure 3. In the following section we look at phenomenological extensions and will focus on charmonium where it is expected that our model will be most applicable.

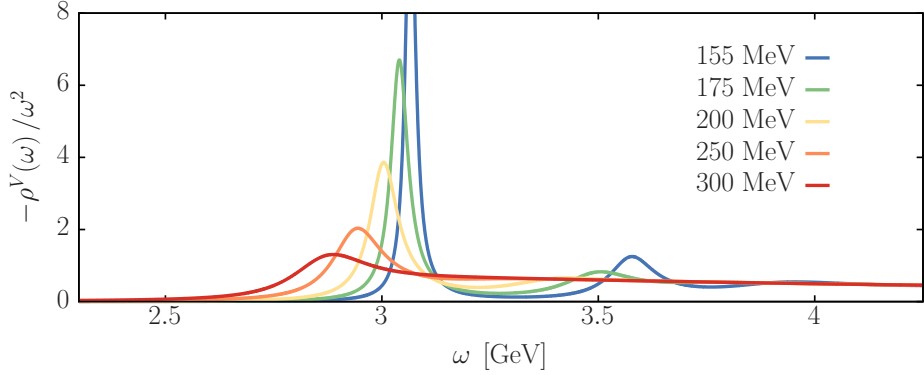

**Figure 2.** Illustrative spectral functions for S-wave Charmonium.

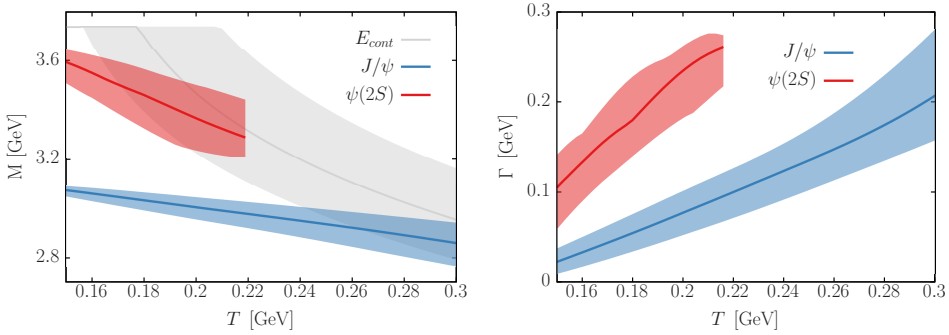

**Figure 3.** Thermal mass (**left**) and spectral width (**right**) of charmonium as a function of temperature. The error bands denote the Debye mass uncertainty arising from the fitting procedure. The continuum threshold energy on the left figure is defined as $\mathrm{Re}V(r \to \infty)$.

### 3.2. Applications to Heavy Ion Collisions

An observable of current interest at RHIC and LHC is the production ratio of $\Psi'$ to $J/\Psi$ particles. The reason is that it is expected to be highly discriminatory among different phenomenological models. Using thermal in-medium quarkonium spectral functions this ratio has already been estimated at vanishing baryo-chemical potential in [13], showing good agreement with predicitons from the statistical model of hadronisation. Here we wish to extend the computation of the ratio to different (lower) beam energies, relevant for future collider facilities such as FAIR and NICA.

We require a prescription to evaluate our Gauss law potential model at a given centre-of-mass energy. The strategy here is two-fold. Firstly, we note that the statistical hadronisation model already provides a well-established scheme with which to estimate the thermal parameters (temperature and

baryo-chemical potential $\mu_B$) of the produced bulk medium at chemical freeze-out with a given $\sqrt{s_{NN}}$. The most recent results [15] are:

$$T(\sqrt{s_{NN}}) = \frac{158 \text{ MeV}}{1 + \exp(2.60 - \ln(\sqrt{s_{NN}})/0.45)}, \qquad \mu_B(\sqrt{s_{NN}}) = \frac{1307.5 \text{ MeV}}{1 + 0.288\sqrt{s_{NN}}}, \qquad (20)$$

where $\sqrt{s_{NN}}$ is the dimensionless numerical value of the centre-of-mass energy measured in GeV.

Secondly, since the physical information within our potential model is captured entirely by the dependence on the Debye mass $m_D$, we need only modify $m_D$ to include the effects on finite baryo-chemical potential. At leading order, the Debye mass can be calculated perturbatively at finite baryo-chemical potential [16]. As a first step, we propose to add this $\mu_B$-term to the temperature dependence of the Debye mass in Equation (19). The result is:

$$m_D(T, \mu_B) = \sqrt{m_D(T, 0)^2 + T^2 g(\Lambda)^2 \frac{N_f}{18\pi^2} \frac{\mu_B^2}{T^2}}. \qquad (21)$$

Here, the renormalisation scale is now $\Lambda = 2\pi\sqrt{T^2 + \mu_B^2/\pi^2}$. At high $\mu_B$ the chemical potential itself becomes the only relevant scale and a similar (linear) dependence of $m_D$ is expected. This leads us to adopt Equation (21) over the entire finite baryo-chemical potential regime. In the absence of reliable lattice data at finite chemical potential, we hold the non-perturbative constants $\kappa_1$ and $\kappa_2$ in Equation (19) the same.

With all ingredients now in place, we may now compute the compute the $\Psi'/J/\Psi$ ratio over a range of centre-of-mass energies. Through Equations (21) and (20) we scan the $\sqrt{s_{NN}}$ range and update the Debye mass that encodes the physics of our potential model. The in-medium spectral functions are calculated in the same manner as Section 3.1 and finally, the number ratio is estimated via the procedure in [13]—assuming an instantaneous freeze-out scenario where all in-medium bound states are projected onto the corresponding vacuum state. The final ratio is expressed as

$$\left.\frac{N_{\Psi'}}{N_{J/\Psi}}\right|_{\sqrt{s_{NN}}} = \left.\frac{R_{\ell\ell}^{\Psi'}}{R_{\ell\ell}^{J/\Psi}}\right|_{\sqrt{s_{NN}}} \times \frac{M_{\Psi'}^2 |\psi_{J/\Psi}(0)|^2}{M_{J/\Psi}^2 |\psi_{\Psi'}(0)|^2}, \quad R_{\ell\ell}^{\Psi_n} \propto A_n \int d^3\mathbf{p}\, n_B\left(\sqrt{M_n^2 + \mathbf{p}^2}\right) \frac{M_n}{\sqrt{M_n + \mathbf{p}^2}}. \qquad (22)$$

Here, $M_n$ is the thermal mass of the state, i.e., the frequency at which the corresponding spectral peak occurs and $A_n$ is the area underneath the peak. The second factor on the right-hand-side of Equation (22) is the square of the $T = 0$ wavefunction at $r = 0$ divided by the square of the mass of each state and is required to obtain the total number density from $R_{\ell\ell}^{\Psi_n}$, which only includes electromagnetic decays [17].

The final results from this entire procedure are plotted in Figure 4, together with the prediction by the statistical hadronisation model. Our analysis shows very good agreement with both the statistical model and the latest experimental results, strengthening the interpretation that charm quarks thermalise before reaching the freeze-out boundary.

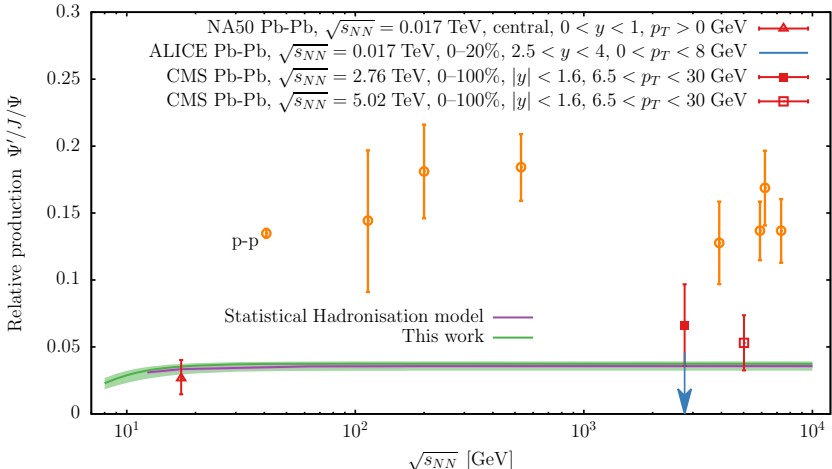

**Figure 4.** The prediction of this work (green) for the relative production yield of $\Psi'$ to $J/\Psi$. We also include the statistical hadronisation model prediction [15] (purple) and experimental data measured by the NA50 [18], ALICE [19], and CMS [20,21] collaborations (red) for Pb–Pb collisions, as well as the pp baseline [15,22] (orange).

## 4. Conclusions

We have presented an improved parametrisation of the in-medium heavy quark potential by employing a generalised Gauss law ansatz in linear response theory. The resulting analytic expressions depended only on a single temperature dependent parameter and were able to quantitatively reproduce the lattice results for the real part of the potential. The resulting imaginary part showed an unphysical logarithmic divergence which we attributed to the equally unphysical unending linear rise of the vacuum Cornell potential. By regularising this artefact, we were able to give physically sound predictions for the imaginary part that in turn qualitatively matched the lattice data. Furthermore, our prescription can be easily extended to model a finite baryo-chemical potential, a region currently inaccessible to lattice QCD simulations. Using the values for $\mu_B$ obtained in the statistical model of hadronisation we computed $\Psi'$ to $J/\Psi$ production yield ratio for different beam energies. The extension of the Gauss-law parametrisation to finite velocity remains work in progress.

**Author Contributions:** conceptualization, A.R.; formal analysis, D.L.; writing–original draft preparation, D.L.; writing–review and editing, A.R.

**Funding:** This study is part of and supported by the DFG Collaborative Research Centre "SFB 1225 (ISOQUANT)"

**Acknowledgments:** We are grateful to Anton Andronic for providing the statistical hadronisation model results.

**Conflicts of Interest:** The authors declare no conflict of interest.

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
