# Peer review of "Quarkonium Phenomenology from a Generalised Gauss Law"

_universe, doi:10.3390/universe5050119_

Reviewer 1 Report

This paper presents an interesting new study of the heavy quark potential in medium. The results are clearly explained and will be of interest to experiments with heavy ions at the LHC.

The paper can be published without revision.

Author Response

We thank the referee for his/her reading of the manuscript. 

Reviewer 2 Report

see attached

Reviewer 3 Report

The authors present a phenomenological model of charmonium production in heavy ion collisions.  The model has the ability to reproduce several important features derived from lattice QCD, which provides an important check to the self-consistency of the model.  The methods are quite well described in the document.  I only have a few minor comments that the authors may wish to address :

Figs. 2 and 3 show results pertaining to bottomonium suppression.  However, the only comparison with experimental results is from the charmonium sector.  Why are there no comparisons with, for example, CMS data in upsilon suppression in PbPb?  Does the lack of a parametrization for finite velocity prohibit this?  Some more explanation of the differences between charm and bottom calculations would be helpful to the reader.

There have been multiple observations of psi(2S)/J/psi suppression in small system at RHIC and the LHC, c.f. ALICE J. High Energ. Phys. (2014) 2014:73 and PHENIX. Phys. Rev. C 95, 034904 (2017).  The possibility of plasma formation in small systems is one of the most debated points in heavy ion physics today, so the authors may wish to comment on these results and how their model could help quantify various effects in heavy ion collisions, such as suppression in plasma (which is calculated here) versus late stage breakup.

Fig 4:  The color difference between the red and brown points is hard to see on the copy of the paper I printed out.  You may wish to change one of the colors to black, or change the marker style, so the reader can more easily tell the difference between the pp and PbPb results.

Author Response

We thank the referee for his/her useful comments which we address below:

1. As the focus of this proceeding lies on charmonium physics, we have replaced Fig. 2 & 3 originally showing in-medium bottomonium properties with the corresponding charmonium ones and thank the referee for pointing out the potential for confusion here. The reason to stay within the charmonium family is that measurements of J/Psi elliptic flow have confirmed that charm quarks are at least in partical kinetic equilibrium with the environment, so that a purely thermal ansatz, as made in our study, may be appropriate. For bottomonium, where no direct evidence of thermalisation has been obtained yet, it is questionable to start out from a fully thermal scenario. In addition, as the referee pointed out, high p_T and non-vanishing rapidity windows at CMS would require a fintie velocity description, which is work in progress.

2. A similar reasoning prevents us from deploying our model to small systems. The starting point of the in-medium spectral functions here is a static and infintely extended medium surrounding the heavy quark pair. While already an idealization for A+A collisions we believe that in small systems are more realistic approach will be necessary.

3. We have changed the colours in Fig. 4.

Round  2

Reviewer 2 Report

This is a paper by PHENIX with the psi' to J/psi ratio, at least in p+p and p+A collisions that the authors could cite.

I am satisfied with the author's reply otherwise.  The manuscript can now be accepted.Relative Yields and Nuclear Modification of ψ′ to J /ψ mesons in p+p, p(3He)+A Collisions  at sqrt s = 200 GeV , measured in PHENIX
PHENIX Collaboration (Axel Drees for the collaboration). 2017.  4 pp. Published in Nucl.Part.Phys.Proc. 289-290 (2017) 417-420. DOI: 10.1016/j.nuclphysbps.2017.05.097. Conference: C16-09-23Proceedings

Author Response

We thank the referee for a second reading of the manuscript.

We have updated Fig. 4 to include the pp results from PHENIX.